# PolyNet: Learning Diverse Solution Strategies for Neural Combinatorial Optimization

## Abstract

In recent years, learning-based approaches have made remarkable strides in tackling combinatorial optimization problems. Reinforcement learning-based construction methods, in particular, have shown promise in producing high-quality solutions, often surpassing established operations research heuristics for simple routing problems. Nonetheless, inherent limitations, such as a lack of solution diversity and limited applicability to complex problems, have hindered their widespread adoption. This paper introduces PolyNet, a novel approach that uses a single-decoder model to learn complementary solution strategies for combinatorial optimization problems, allowing the rapid creation of diverse solutions for a given instance. Moreover, PolyNet's diversity mechanism enhances training exploration without relying on solution space symmetries, enabling it to effectively tackle more complex problems. We evaluate PolyNet on three combinatorial optimization problems of varying difficulty. Our comprehensive experiments consistently demonstrate significant improvements over state-of-the-art machine learning methods, both in terms of swift solution generation and extensive search.

## 1 Introduction

In recent years, there have been remarkable advancements in the field of learning-based approaches for solving combinatorial optimization (CO) problems (Bello et al., 2016; Kool et al., 2019; Kwon et al., 2020). Notably, reinforcement learning (RL) methods have emerged that build a solution to a problem step-by-step in a sequential decision making process. Initially, these construction techniques struggled to produce high-quality solutions. However, recent methods have surpassed even established operations research heuristics, such as LKH3, for simpler, smaller-scale routing problems. Learning-based approaches thus now have the potential to become versatile tools, capable of learning specialized heuristics tailored to unique business-specific problems. Moreover, with access to sufficiently large training datasets, they may consistently outperform off-the-shelf solvers in numerous scenarios. This work aims to tackle some of the remaining challenges that currently impede the widespread adoption of learning-based heuristic methods in practical applications.

A key limitation of learning-based approaches is that they often struggle to produce *diverse* solutions, leading to diminishing returns when generating more than a few hundred solutions per instance, as noted in Grinsztajn et al. (2022). This limitation is especially problematic when decision makers are willing to accept longer runtimes for better results. To address this issue, Xin et al. (2021) propose a transformer model with multiple decoders that encourage distinct solution strategies during training by maximizing the Kullback-Leibler divergence between decoder output probabilities. However, to manage computational costs, diversity is only promoted in the initial construction step. In contrast, Grinsztajn et al. (2022) introduce Poppy, a training procedure for multi-decoder models that increases diversity without relying on Kullback-Leibler divergence. Poppy specializes a population (i.e., a set) of decoders during the learning phase by training only the best-performing decoder for each problem instance. While effective, Poppy is computationally intensive, requiring a separate decoder for each policy, thus limiting the number learnable policies per problem.

Another significant limitation of existing neural CO approaches is their focus on relatively simple problems with a limited number of constraints. For example, many recent publications have made considerable progress solving the traveling salesperson problem (TSP) (e.g., Jin et al. (2023); Xiao et al. (2023)). However, the majority of real-world optimization problems are significantly more

complex than the TSP, involving a greater number of constraints that must be satisfied. Furthermore, approaches tailored exclusively to the TSP are unlikely to be effective for these more complex problems. Even techniques evaluated on both the TSP and the significantly more complex capacitated vehicle routing problem (CVRP) do not seamlessly extend to other routing problems. For example, many recent neural construction methods (e.g., Li et al. (2023); Choo et al. (2022)) build on the POMO approach (Kwon et al., 2020) that enhances exploration during training by forcing diverse first actions during solution construction. This assumes that the first construction action has a negligible impact on solution quality, which is true for the TSP and CVRP due to symmetries in the solution space. However, in more complex optimization problems the initial action often significantly influences solution quality, rendering these methods less effective at generating solutions.

In this paper, we introduce PolyNet, which effectively addresses the previously discussed limitations through the following innovations:

1. PolyNet learns a diverse and complementary set of solution strategies for optimization problems using a single decoder, a departure from existing diversity-focused approaches.
2. PolyNet eliminates the requirement to enforce the first construction action, broadening its applicability to a wider range of CO problems.

By utilizing a single decoder to learn multiple strategies, PolyNet allows to quickly generate a set of diverse solutions for a problem instance. This significantly enhances exploration, allowing us to find better solution during training and testing. Furthermore, by abandoning the concept of forcing diverse first actions, we exclusively rely on PolyNet's inherent diversity mechanism to facilitate exploration during the search process. This fundamental change not only enhances the method's adaptability to new problems but also leads to substantial performance enhancements, particularly in solving complex problems such as the capacitated vehicle routing problem (CVRP).

We assess PolyNet's performance across three problems: the TSP, the CVRP, and the CVRP with time windows (CVRPTW), each involving instances with up to 300 nodes. We search for high-quality solutions with PolyNet by either quickly sampling a set of diverse solutions or by a synergistic combination of PolyNet with the efficient active search (EAS) technique (Hottung et al., 2022). EAS fine-tunes a subset of model parameters during testing and works exceptionally well with PolyNet. This synergy arises from PolyNet's capability to generate diverse solutions, which empowers EAS to explore a broader spectrum of potential solutions during the search process. Across all problems, PolyNet consistently demonstrates a significant advancement over the state-of-the-art in both swift solution generation and comprehensive search efforts.

## 2 LITERATURE REVIEW

**Neural CO** In their seminal work, Vinyals et al. (2015) introduce the novel pointer network architecture, an early application of modern machine learning methods to solve CO problems. Pointer networks autoregressively generate discrete outputs corresponding to input positions. When trained via supervised learning, they can produce near-optimal solutions for the TSP with up to $50$ nodes. Bello et al. (2016) propose training pointer networks using reinforcement learning instead and illustrate the efficacy of this method in solving larger instances of the TSP.

Nazari et al. (2018) use the pointer network architecture to solve the CVRP with 100 nodes. Kool et al. (2019) improve on this architecture by utilizing a transformer-based encoder with self-attention (Vaswani et al., 2017). Recognizing that many CO problems contain symmetries, Kwon et al. (2020) propose POMO, a method that leverages such symmetries to generate better solutions. Kim et al. (2022) extend these ideas and propose a general-purpose symmetric learning scheme. Drakulic et al. (2023) use bisimulation quotienting (Givan et al., 2003) to improve out-of-distribution generalization of neural CO methods. Only few works, including (Falkner & Schmidt-Thieme, 2020; Kool et al., 2022a), propose neural CO approaches for routing problems with time windows.

Instead of constructing solutions autoregressively, some methods predict a heat-map that outlines promising edges. This is subsequently used in a post-hoc search for solution construction (Joshi et al., 2019; Fu et al., 2021; Kool et al., 2022b). Another class of methods iteratively improves initial yet complete solutions. For instance, Hottung & Tierney (2020) propose a framework that iteratively destroys parts of a solution using handcrafted procedures and then repairs it using a learned

operator to explore the solution space. Similarly, Ma et al. (2021) learn to iteratively improve an initial solution by performing local adjustments, while Chen & Tian (2019) pose CO as a sequential rewriting problem in which parts of a solution are iteratively changed using a learned routine.

Hottung et al. (2022) introduce EAS, a method that guides the search by updating a subset of the policy parameters during inference. Similarly, Choo et al. (2022) propose SGBS, a post-hoc inference mechanism that combines Monte-Carlo tree search with beam search to provide search guidance. When used in combination with EAS, it achieves state-of-the-art performance on several problems.

**Diversity mechanisms in RL** In RL, skill-learning algorithms aim to discover a set of policies with diverse behaviors (defined by the visited states) to accelerate task-specific training. For instance, Eysenbach et al. (2018) and Sharma et al. (2019) learn skills that exhibit predictable behavior and are as diverse as possible. Here, a skill corresponds to the policy conditioned on some latent context.

In contrast to implicitly maintaining a collection of agents through context, population-based RL techniques explicitly maintain a finite agent population and use diversity mechanisms to discover diverse strategies for solving RL tasks. The population could be constructed iteratively. For example, Zhang et al. (2019) learn a collection of policies that solve a task using distinct action sequences. Alternatively, agents can be trained in a population-based setup (PIERROT & Flajolet, 2023). Recently, Wu et al. (2023) focus on task-specific diversity defined according to user-specified behavior descriptors and employ population-based training to maximize diversity at different quality levels.

Diversity mechanisms are also used in single and multi-agent setups to learn diverse problem-solving strategies. For instance, in neural program synthesis, Bunel et al. (2018) optimize the expected reward when sampling a pool of solutions and keeping the best one. This encourages the policy to diversify its choices by assigning probability mass to several solutions. In multi-agent RL, Li et al. (2021) use the mutual information between agents' identities and trajectories as an intrinsic reward to promote diversity, and solve cooperative tasks requiring diverse strategies among the agents.

**Diversity mechanisms in neural CO** Kim et al. (2021) present a hierarchical strategy for solving routing problems, where a learned seeder policy maximizes solution diversity through entropy rewards and generates diverse candidate solutions. These are then modified by a learned reviser policy to improve solution quality. As previously discussed, Xin et al. (2021) encourage diverse solutions using multiple decoders and KL-divergence regularization, while Grinsztajn et al. (2022) use a population of agents through multiple decoders to learn complementary strategies, updating exclusively the best-performing agent at each iteration. This objective shares similarities with (Bunel et al., 2018), in which also only the best solution is considered.

## 3 POLYNET

### 3.1 BACKGROUND

Neural CO approaches seek to train a neural network denoted as $\pi_\theta$ with learnable weights $\theta$. The network's purpose is to generate a solution $\tau$ when provided with an instance $l$. To achieve this, we employ RL techniques and model the problem as a Markov decision process (MDP), wherein a solution is sequentially constructed in $T$ discrete time steps. At each step $t \in (1, \ldots, T)$, an action $a_t$ is selected based on the probability distribution $\pi_\theta(a_t|s_t)$ defined by the neural network where $s_t$ is the current state. The initial state $s_1$ encapsulates the information about the problem instance $l$, while subsequent states $s_{t+1}$ are derived by applying the action $a_t$ to the previous state $s_t$. A (partial) solution denoted as $\bar{\tau}_t$ is defined by the sequence of selected actions $(a_1, a_2, \ldots, a_t)$. Once a complete solution $\tau = \bar{\tau}_T$ satisfying all problem constraints is constructed, we can compute its associated reward $\mathcal{R}(\tau, l)$. The overall policy of generating a solution $\tau$ for an instance $l$ is defined as $\pi_\theta(\tau \mid l) = \prod_{t=1}^{T} \pi_\theta(a_t \mid s_t)$.

### 3.2 OVERVIEW

PolyNet is a learning-based approach designed to learn a set of diverse solution strategies for CO problems. During training, each strategy is allowed to specialize on a subset of the training data, and thus need not be the best strategy for the entire dataset. This essentially results in a portfolio of strategies, which are known to be highly effective for solving CO problems (Bischl et al., 2016).

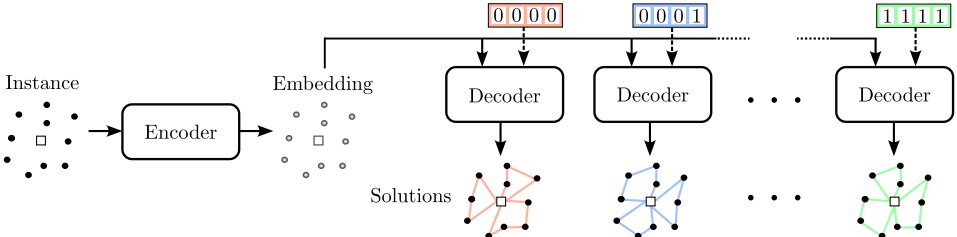

Figure 1: PolyNet solution generation.

Our pursuit of diversity is fundamentally a means to enhance exploration and consequently solution quality. Note that PolyNet not only enhances performance at test time (where we sample multiple solutions for each strategy and keep only the best one), but also improves exploration during training.

PolyNet aims to learn $K$ different solution strategies $\pi_1, \ldots, \pi_K$ using a single neural network. To achieve this, we condition the solution generation process on an additional input $v_i \in \{v_1, \ldots, v_K\}$ that defines which of the strategies should be used to sample a solution so that

$$\pi_1, \ldots, \pi_K = \pi_\theta(\tau_1 \mid l, v_1), \ldots, \pi_\theta(\tau_K \mid l, v_K). \tag{1}$$

We use a set of unique bit vectors for $\{v_1, \ldots, v_K\}$. Alternative representations should also be feasible as long as they are easily distinguishable by the network.

PolyNet uses a neural network that builds on the established transformer architecture for neural CO (Kool et al., 2019). The model consists of an encoder that creates an embedding of a problem instance, and a decoder that generates multiple solutions for an instance based on the embedding. To generate solutions quickly, we only insert the bit vector $v$ into the decoder, allowing us to generate multiple diverse solutions for an instance with only a single pass through the computationally expensive encoder. Figure 1 shows the overall solution generation process of the model where bit vectors of size 4 are used to generate to generate $K = 16$ different solutions for a CVRP instance.

### 3.3 TRAINING

During training we (repeatedly) sample $K$ solutions $\{\tau_1, \ldots, \tau_K\}$ for an instance $l$ based on $K$ different vectors $\{v_1, \ldots, v_K\}$, where the solution $\tau_i$ is sampled from the probability distribution $\pi_\theta(\tau_i \mid l, v_i)$. To allow the network to learn $K$ different solution strategies, we follow the Poppy method (Grinsztajn et al., 2022) and only update the model weights with respect to the best of the $K$ solutions. Let $\tau^*$ be the best solution, i.e., $\tau^* = \arg\min_{\tau_i \in \{\tau_1, \ldots, \tau_K\}} \mathcal{R}(\tau_i, l)$, and let $v^*$ be the corresponding vector (ties are broken arbitrarily). We then update the model using the gradient

$$\nabla_\theta \mathcal{L} = \mathbb{E}_{\tau^*} \big[ (R(\tau^*, l) - b_\circ) \nabla_\theta \log \pi_\theta(\tau^* \mid l, v^*) \big], \tag{2}$$

where $b_\circ$ is a baseline (which we discuss in detail below). Updating the model weights only based on the best found solution allows the network to learn specialized strategies that do not have to work well on all instances from the training set. While this approach does not explicitly enforce diversity, it incentivizes the model to learn diverse strategies in order to optimize overall performance. We show experimentally that this is the case. Grinsztajn et al. (2022) discuss this loss in more detail.

**Exploration & Baseline** Most recent neural construction heuristics follow the POMO approach and rollout $N$ solutions from $N$ different starting nodes per instance to increase exploration. This is possible because many CO problems contain symmetries in the solution space that allow an optimal solution to be found from all $N$ starting nodes. In practice, this mechanism is implemented by forcing a different first construction action for each of the $N$ rollouts. Forcing diverse rollouts not only improves exploration, but also allows the average reward of all $N$ rollouts to be used as a baseline. However, this exploration mechanism should not be used when the first action can not be freely chosen without impacting the solution quality.

In PolyNet, we do not use an exploration mechanism or a baseline that assumes symmetries in the solution space. Instead, we only rely on the exploration provided by our conditional solution generation. This allows us to solve a wider range of optimization problems. As a baseline we simply use the average reward of all $K$ rollouts for an instance, i.e., $b_\circ = \frac{1}{K} \sum_{i=1}^{K} R(\tau_i, l)$.

### 3.4 NETWORK ARCHITECTURE

PolyNet extends the neural network architecture of the POMO approach by a new residual block in the decoder. This design allows us to start PolyNet's training from an already trained POMO model, which significantly reduces the amount of training time needed. Figure 2 shows the overall architecture of the modified decoder including the new PolyNet layers. The new layers accept the bit vector $v$ as input and use it to calculate an update for the output of the masked multi-head attention mechanism. They consist of a concatenation operation followed by two linear layers with a ReLU activation function in between. See Kwon et al. (2020) for more details on the encoder and decoder.

The output of the new layers directly impacts the final pointer mechanism that is used to calculate a probability distribution over all available actions, allowing the new layers to significantly influence the solution generation based on $v$. However, the model can also learn to completely ignore $v$ by setting all weights of the second linear layer to zero. This is intentional, as our objective is to increase diversity via the loss function, rather than force unproductive diversity through the network architecture. Additionally, it's worth noting that the dimensions of the linear layers naturally constrain the achievable diversity within the learned strategies.

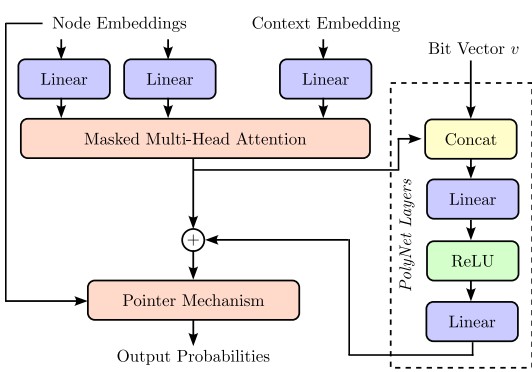

Figure 2: Decoder architecture.

### 3.5 SEARCH

A simple and fast search procedure given an unseen test instance $l$ is to sample multiple solutions in parallel and return the best one. Specifically, to construct a set of $M$ distinct solutions, we initially draw $M$ binary vectors from the set $v_1, \ldots, v_K$, allowing for replacement if $M$ exceeds the size of $K$. Subsequently, we employ each of these $M$ vectors to sample individual solutions. This approach generates a diverse set of instances in a parallel and independent manner, making it particularly suitable for real-world decision support settings where little time is available.

To facilitate a more extensive, guided search, we combine PolyNet with EAS. EAS is a simple technique to fine-tune a subset of model parameters for a single instance in an iterative process driven by gradient descent. In contrast to the EAS variants described in Hottung et al. (2022), we do not insert any new layers into the network or update the instance embeddings. Instead, we only fine-tune the new PolyNet layers during search. Since PolyNet is specifically trained to create diverse solutions based on these layers, EAS can easily explore a wide variety of solutions during the search while modifying only a portion of the model's parameters.

## 4 EXPERIMENTS

We compare PolyNet's search performance to state-of-the-art methods on three established problems. Additionally, we explore solution diversity during the training and testing phases and analyze the impact of the free first move selection. Our experiments are conducted on a GPU cluster utilizing a single Nvidia A100 GPU per run. In all experiments, the new PolyNet layers comprise two linear layers, each with a dimensionality of 256. PolyNet will be made publicly available upon acceptance.

### 4.1 PROBLEMS

**TSP**   The TSP is a thoroughly researched routing problem in which the goal is to find the shortest tour among a set of $n$ nodes. The tour must visit each node exactly once and then return to the initial starting node. We consider the TSP due to its significant attention from the machine learning (ML) community, making it a well-established benchmark for ML-based optimization approaches. However, it is important to note that instances with $n \leq 300$ can be quickly solved to optimality by CO solvers that have been available for many years. To generate problem instances, we adhere to the methodology outlined in Kool et al. (2019).

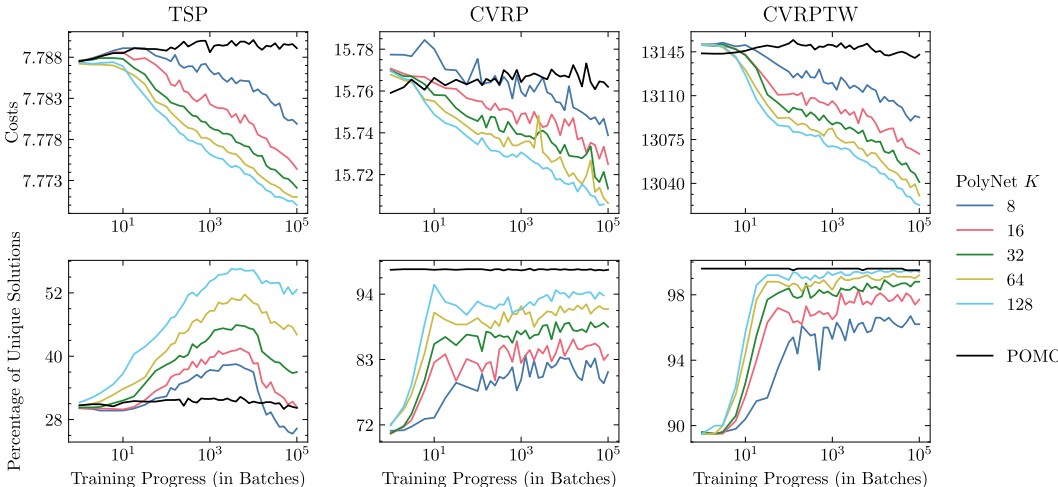

Figure 3: Validation performance during training (log scale).

**CVRP** The objective of the CVRP is to determine the shortest routes for a fleet of vehicles tasked with delivering goods to a set of $n$ customers. These vehicles start and conclude their routes at a central depot and have a limited capacity of goods that they can carry. CVRP instances are considerably more difficult to solve than TSP instances of equivalent size (despite both problems being of the same computational complexity class). Even cutting-edge CO methods struggle to reliably find optimal solutions for instances of moderate size with $n \leq 300$ customers. To generate problem instances, we again adopt the approach outlined in Kool et al. (2019), where we assign vehicle capacities of 50, 60, 70, and 80 for instances involving 100, 150, 200, and 300 customers, respectively.

**CVRPTW** The CVRPTW is an extension of the CVRP that introduces temporal constraints limiting when a customer can receive deliveries from a vehicle. Each customer $i$ is associated with a time window, comprising an earliest arrival time $e_i$ and a latest allowable arrival time $l_i$. Vehicles may arrive early at a customer $i$, but they must wait until the specified earliest arrival time $e_i$ before making a delivery. The travel duration between customer $i$ and customer $j$ is calculated as the Euclidean distance between their locations, and each delivery has a fixed duration. All vehicles initiate their routes from the depot at time 0. We use the CVRP instance generator outlined in Queiroga et al. (2021) to generate customer positions and demands, and we adhere to the methodology established in Solomon (1987) for generating the time windows. It's essential to note that customer positions are not sampled from a uniform distribution; instead, they are clustered to replicate real-world scenarios. Further details for instance generation for the CVRPTW are given in Appendix C.

## 4.2 TRAINING

To assess the effectiveness of our diversity mechanism, we conduct short training runs of PolyNet with varying values of $K$. As a baseline we report results for the training of the POMO model. For all runs, we use a batch size of 480 and a learning rate set at $10^{-4}$. To ensure a stable initial state for training runs, we start all runs from a trained POMO model. For PolyNet, we initialize the PolyNet layer weights to zero, minimizing initial randomness. During training, we perform regular evaluations on a separate validation set of $10,000$ instances, sampling 800 solutions per instance.

Our evaluation tracks the cost of the best solution and the percentage of unique solutions among the 800 solutions per instance. Figure 3 presents the evaluation results. Across all three problems, we observe a clear trend: higher values of $K$ lead to a more rapid reduction in average costs on the validation set and are associated with a greater percentage of unique solutions. Note that POMO does not benefit from further training. These results underscore the effectiveness of our approach in promoting solution diversity and improving solution quality.

Table 1: Search performance results for TSP.

| | Method | Test (10K instances) $n_{tr}=n_{eval}=100$ | | | Test (1K instances) $n_{tr}=n_{eval}=200$ | | | Generalization (1K instances) $n_{tr}=100, n_{eval}=150$ | | | $n_{tr}=200, n_{eval}=300$ | | |
|---|---|---|---|---|---|---|---|---|---|---|---|---|---|
| | | Obj. | Gap | Time | Obj. | Gap | Time | Obj. | Gap | Time | Obj. | Gap | Time |
| | Concorde | 7.765 | - | 82m | 10.687 | - | 31m | 9.346 | - | 17m | 12.949 | - | 83m |
| | LKH3 | 7.765 | 0.000% | 8h | 10.687 | 0.000% | 3h | 9.346 | 0.000% | 99m | 12.949 | 0.000% | 5h |
| Unguided | POMO greedy | 7.775 | 0.135% | 1m | 10.770 | 0.780% | <1m | 9.393 | 0.494% | <1m | 13.216 | 2.061% | 1m |
| | sampling | 7.772 | 0.100% | 3m | 10.759 | 0.674% | 2m | 9.385 | 0.411% | 1m | 13.257 | 2.378% | 7m |
| | Poppy | 7.766 | 0.015% | 4m | 10.711 | 0.226% | 2m | 9.362 | 0.164% | 1m | 13.052 | 0.793% | 7m |
| | PolyNet | 7.765 | **0.000%** | 4m | 10.690 | **0.032%** | 2m | 9.352 | **0.055%** | 1m | 12.995 | **0.351%** | 8m |
| Guided | DPDP | 7.765 | 0.004% | 2h | - | - | - | 9.434 | 0.937% | 44m | - | - | - |
| | MDAM | 7.781 | 0.208% | 4h | - | - | - | 9.403 | 0.603% | 1h | - | - | - |
| | POMO EAS | 7.769 | 0.053% | 3h | 10.720 | 0.310% | 3h | 9.363 | 0.172% | 1h | 13.048 | 0.761% | 8h |
| | SGBS | 7.769 | 0.058% | 9m | 10.727 | 0.380% | 24m | 9.367 | 0.220% | 8m | 13.073 | 0.951% | 77m |
| | SGBS+EAS | 7.767 | 0.035% | 3h | 10.719 | 0.300% | 3h | 9.359 | 0.136% | 1h | 13.050 | 0.776% | 8h |
| | PolyNet EAS | 7.765 | **0.000%** | 3h | 10.687 | **0.001%** | 2h | 9.347 | **0.001%** | 1h | 12.952 | **0.018%** | 7h |

## 4.3 SEARCH PERFORMANCE

We conduct an extensive evaluation of PolyNet's search performance, benchmarking it against state-of-the-art neural CO methods. To this end, we train separate models for problem instances of size 100 and 200, and then evaluate the models trained on $n=100$ using instances with 100 and 150 nodes, and the models trained on $n=200$ using instances with 200 and 300 nodes. We can thus assess the model's capability to generalize to instances that diverge from the training data. Throughout our evaluation, we employ the instance augmentation technique introduced in Kwon et al. (2020).

For the training of PolyNet models, we set the parameter $K$ to 128 across all problems. For instances of size $n=100$, we train our models for 300 epochs (200 for the TSP), with each epoch comprising $4 \times 10^8$ solution rollouts. For instances with $n=200$, we start training based on the $n=100$ models, running 40 additional training epochs (20 for the TSP). To optimize GPU memory utilization, we adjust the batch size separately for each problem and its dimensions.

We categorize the evaluated algorithms into two groups: unguided and guided methods. Unguided algorithms generate solutions independently, while guided methods incorporate a high-level search component capable of navigating the search space. For a comparison to unguided algorithms, we compare PolyNet to POMO and the Poppy approach with a population size of 8. To ensure fairness we retrain Poppy using the same training setup as for Poppy. Note that POMO has already been trained to full convergence and does not benefit from additional training (see Figure 3). For all approaches, we sample $64 \times n$ solutions per instance (except for POMO using greedy solution generation). For our comparison to guided algorithms, we use PolyNet with EAS and compare it with POMO combined with EAS (Hottung et al., 2022) and SGBS (Choo et al., 2022). In the case of PolyNet, we sample $200 \times 8 \times n$ solutions per instance over the course of 200 iterations. Furthermore, we compare to some problem-specific approaches that are explained below. Note that we provide additional search trajectory plots in Appendix B.

**TSP** We use the 10,000 test instances with $n=100$ from Kool et al. (2019) and test sets consisting of $1,000$ instances from Hottung et al. (2021) for $n=150$ and $n=200$. For $n=300$, we generate new instances. As a baseline, we use the exact solver Concorde (Applegate et al., 2006) and the heuristic solver LKH3 (Helsgaun, 2017). Additionally, we also compare to DPDP (Kool et al., 2022b) and the diversity-focused method MDAM (Xin et al., 2021) with a beam search width of 256.

Table 1 provides our results on the TSP, showing clear performance improvements of PolyNet during fast solution generation and extensive search with EAS for all considered instance sets. For TSP instances with 100 nodes, PolyNet achieves a gap that is practically zero while being roughly 120 times faster than LKH3. Furthermore, on all four instance sets, PolyNet with unguided solution sampling finds solutions with significantly lower costs in comparison to guided learning approaches while reducing the runtime by a factor of more than 100 in many cases.

**CVRP** Similar to the TSP, we use the test sets from Kool et al. (2019) and Hottung et al. (2021). As a baseline, we use LKH3 (Helsgaun, 2000) and the state-of-the-art (OR) solver HGS (Vidal et al.,

Table 2: Search performance results for CVRP.

| | Method | Test (10K instances) $n_{tr}=n_{eval}=100$ | | | Test (1K instances) $n_{tr}=n_{eval}=200$ | | | Generalization (1K instances) $n_{tr}=100, n_{eval}=150$ | | | $n_{tr}=200, n_{eval}=300$ | | |
|---|---|---|---|---|---|---|---|---|---|---|---|---|---|
| | | Obj. | Gap | Time | Obj. | Gap | Time | Obj. | Gap | Time | Obj. | Gap | Time |
| | HGS | 15.563 | - | 54h | 21.766 | - | 17h | 19.055 | - | 9h | 27.737 | - | 46h |
| | LKH3 | 15.646 | 0.53% | 6d | 22.003 | 1.09% | 25h | 19.222 | 0.88% | 20h | 28.157 | 1.51% | 34h |
| Unguided | POMO greedy | 15.754 | 1.23% | 1m | 22.194 | 1.97% | <1m | 19.684 | 3.30% | <1m | 28.627 | 3.21% | 1m |
| | sampling | 15.705 | 0.91% | 5m | 22.136 | 1.70% | 3m | 20.109 | 5.53% | 1m | 28.613 | 3.16% | 9m |
| | Poppy | 15.685 | 0.78% | 5m | 22.040 | 1.26% | 3m | 19.578 | 2.74% | 1m | 28.648 | 3.28% | 8m |
| | PolyNet | 15.640 | **0.49%** | 5m | 21.957 | **0.88%** | 3m | 19.501 | **2.34%** | 1m | 28.552 | **2.94%** | 8m |
| Guided | DACT | 15.747 | 1.18% | 22h | - | - | - | 19.594 | 2.83% | 16h | - | - | - |
| | DPDP | 15.627 | 0.41% | 23h | - | - | - | 19.312 | 1.35% | 5h | - | - | - |
| | MDAM | 15.885 | 2.07% | 5h | - | - | - | 19.686 | 3.31% | 1h | - | - | - |
| | POMO EAS | 15.618 | 0.35% | 6h | 21.900 | 0.61% | 3h | 19.205 | 0.79% | 2h | 28.053 | 1.14% | 12h |
| | SGBS | 15.659 | 0.62% | 10m | 22.016 | 1.15% | 7m | 19.426 | 1.95% | 4m | 28.293 | 2.00% | 22m |
| | SGBS+EAS | 15.594 | 0.20% | 6h | 21.866 | 0.46% | 4h | 19.168 | 0.60% | 2h | 28.015 | 1.00% | 12h |
| | PolyNet EAS | 15.584 | **0.14%** | 4h | 21.821 | **0.25%** | 2h | 19.166 | **0.59%** | 1h | 27.993 | **0.92%** | 9h |

Table 3: Search performance results for CVRPTW.

| | Method | Test (10K instances) $n_{tr}=n_{eval}=100$ | | | Test (1K instances) $n_{tr}=n_{eval}=200$ | | | Generalization (1K instances) $n_{tr}=100, n_{eval}=150$ | | | $n_{tr}=200, n_{eval}=300$ | | |
|---|---|---|---|---|---|---|---|---|---|---|---|---|---|
| | | Obj. | Gap | Time | Obj. | Gap | Time | Obj. | Gap | Time | Obj. | Gap | Time |
| | PyVRP | 12,534 | - | 39h | 18,422 | - | 11h | 17,408 | - | 9h | 25,732 | - | 26h |
| Unguided | POMO greedy | 13,120 | 4.67% | 1m | 19,656 | 6.70% | 1m | 18,670 | 7.25% | <1m | 28,022 | 8.90% | 2m |
| | sampling | 13,019 | 3.87% | 7m | 19,531 | 6.02% | 4m | 18,571 | 6.68% | 2m | 28,017 | 8.88% | 13m |
| | Poppy | 12,969 | 3.47% | 5m | 19,406 | 5.34% | 3m | 18,612 | 6.91% | 2m | 28,104 | 9.22% | 10m |
| | PolyNet | 12,876 | **2.73%** | 5m | 19,232 | **4.40%** | 3m | 18,429 | **5.86%** | 2m | 27,807 | **8.07%** | 10m |
| Guided | POMO EAS | 12,762 | 1.81% | 6h | 18,966 | 2.96% | 4h | 17,851 | 2.54% | 2h | 26,608 | 3.40% | 14h |
| | SGBS | 12,897 | 2.89% | 12m | 19,240 | 4.44% | 8m | 18,201 | 4.55% | 4m | 27,264 | 5.95% | 25m |
| | SGBS+EAS | 12,714 | 1.43% | 7h | 18,912 | 2.66% | 4h | 17,835 | 2.45% | 2h | 26,651 | 3.57% | 15h |
| | PolyNet EAS | 12,654 | **0.96%** | 5h | 18,739 | **1.72%** | 3h | 17701 | **1.68%** | 1h | 26,504 | **3.00%** | 10h |

2012; Vidal, 2022). We compare to the same learning methods as for the TSP with the addition of DACT (Ma et al., 2021).

The CVRP results in Table 2 once again indicate consistent improvement across all considered problem sizes. Especially on the instances with 100 and 200 customers, PolyNet improves upon the state-of-the-art learning-based approaches by reducing the gap by more than 30% during fast solution generation and extensive search. Also note that PolyNet significantly outperforms the other diversity-focused approaches Poppy and MDAM.

**CVRPTW** For the CVRPTW, we use the state-of-the-art CO solver PyVRP (Wouda et al., 2023) as a baseline stopping the search after 1,000 iterations without improvement. We compare to a POMO implementation that we adjusted to solve the CVRPTW by extending the node features with the time windows and the context information used at each decoding step by the current time point. These models are trained for 50,000 epochs, mirroring the training setup used for the CVRP.

Table 3 presents the CVRPTW results, demonstrating PolyNet's consistent and superior performance across all settings compared to Poppy and POMO (with SGBS and EAS). Notably, for instances with 100 customers, PolyNet matches almost the CO solver PyVRP with a gap below 1%.

## 4.4 SEARCH DIVERSITY

To evaluate the diversity mechanism of PolyNet, we compare the diversity of solutions generated by PolyNet to those generated by POMO. More specifically, for a subset of test instances, we sample 100 solutions per instance with both approaches and compare the solutions found on the basis of their uniqueness and their cost. We calculate the uniqueness of a solution by using the average of the broken pairs distance (Prins, 2009) to all other 99 generated solutions.

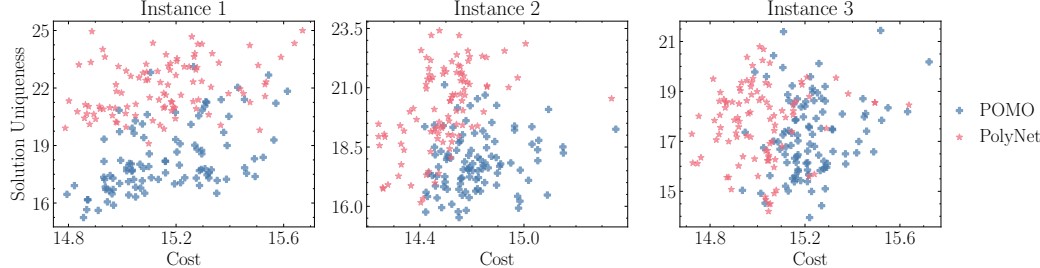

Figure 4: Solution diversity vs. costs for three CVRP instances.

Table 4: Ablation results for free first move selection.

| Method | | TSP | | | CVRP | | | CVRPTW | | |
|---|---|---|---|---|---|---|---|---|---|---|
| | Obj. | Gap | Time | Obj. | Gap | Time | Obj. | Gap | Time |
| PolyNet  Free first move | 7.765 | 0.000% | 4m | 15.640 | 0.49% | 5m | 12,876 | 2.73% | 5m |
| Forced first move | 7.765 | 0.006% | 4m | 15.655 | 0.59% | 5m | 12,909 | 3.00% | 6m |
| Poppy | 7.766 | 0.015% | 4m | 15.685 | 0.78% | 5m | 12,969 | 3.47% | 5m |

Figure 4 shows the results for the first three CVRP instances from the test set. Ideally, we seek solutions with low costs and high uniqueness (i.e., solutions positioned in the top-left corner of the plot). Notably, the majority of solutions laying on the Pareto front of these two objectives originate from PolyNet. In summary, PolyNet is consistently able to generate solutions with higher uniqueness and lower costs, demonstrating its ability to learn strategies that generate not only diverse solutions but also high-quality solutions. See Appendix A for additional visualizations of all three problems.

### 4.5 ABLATION STUDY: FORCING THE FIRST MOVE

In contrast to most state-of-the-art methods, PolyNet does not force diverse first construction actions to enhance exploration during training and testing. Instead, PolyNet relies solely on its built-in diversity mechanism, which does not assume symmetries in the solution space. To assess the efficacy of this approach, we compare the performance of PolyNet with and without forced first move selection when sampling $64 \times 100$ solutions per instance.

Table 4 shows the results for all problems with $n = 100$. Remarkably, across all scenarios, allowing PolyNet to select the first move yields superior performance compared to forcing the first move. This finding is particularly striking for the TSP, where the first move does not affect solution quality. Furthermore, PolyNet with forced first move selection outperforms Poppy (which also enforces the first move), underscoring that PolyNet's single-decoder architecture delivers better results than PolyNet's multi-decoder approach.

## 5 CONCLUSION

We introduced the novel approach PolyNet, which is capable of learning diverse solution strategies using a single-decoder model. PolyNet deviates from the prevailing trend in neural construction methods, in which diverse first construction steps are forced to improve exploration. Instead, it relies on its diverse strategies for exploration, enabling its seamless adaptation to problems where the first move significantly impacts solution quality. In our comprehensive evaluation across three problems, including the more challenging CVRPTW, PolyNet consistently demonstrates performance improvements over all other learning-based methods, particularly those focused on diversity.

Regarding our approach's limitations, we acknowledge that the computational complexity of the attention mechanism we employ restricts its applicability to instances with more than 1,000 nodes. However, it is essential to emphasize that the problem sizes examined in this paper for the CVRP(TW) remain challenging for traditional CO solvers and are highly significant in real-world applications. Furthermore, we note that the black-box nature of the PolyNet's decision-making may be unacceptable in certain decision contexts.

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

## A    ADDITIONAL SOLUTION DIVERSITY VS. COSTS PLOTS

We provide additional plots that show the diversity and costs of solutions sampled with PolyNet and POMO for the TSP, CVRP, and CVRPTW in Figures 5-7. For each problem we report results for the first 6 test set instances. The results further reinforce the notion that PolyNet generates solutions with higher diversity and lower costs than POMO.

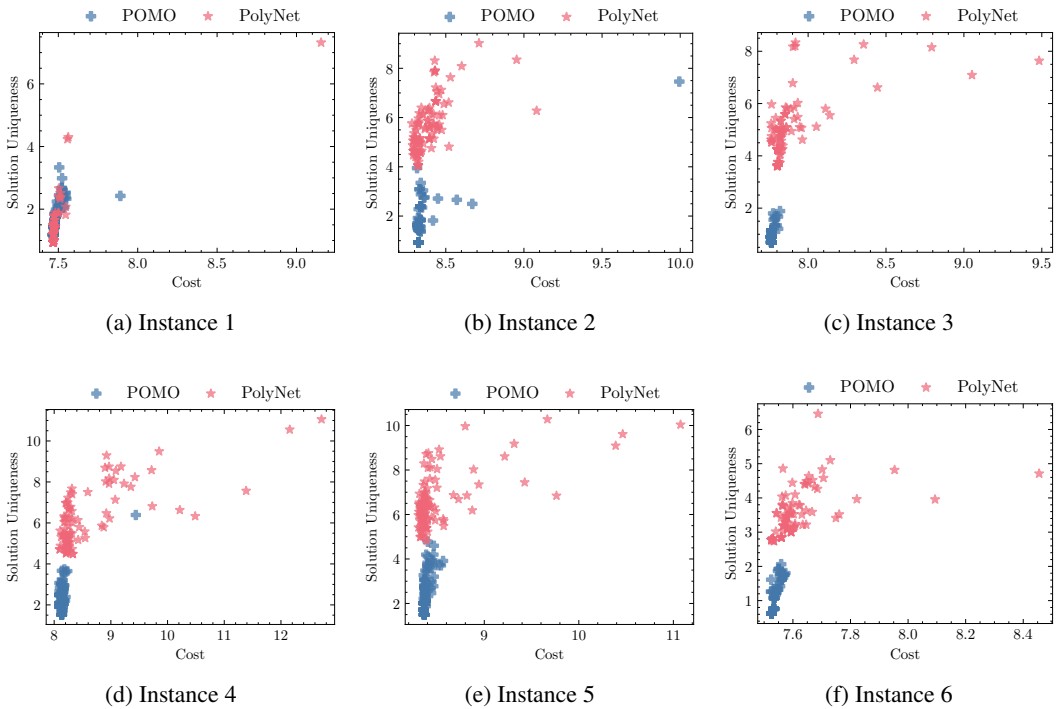

Figure 5: Solution diversity vs. costs for the TSP.

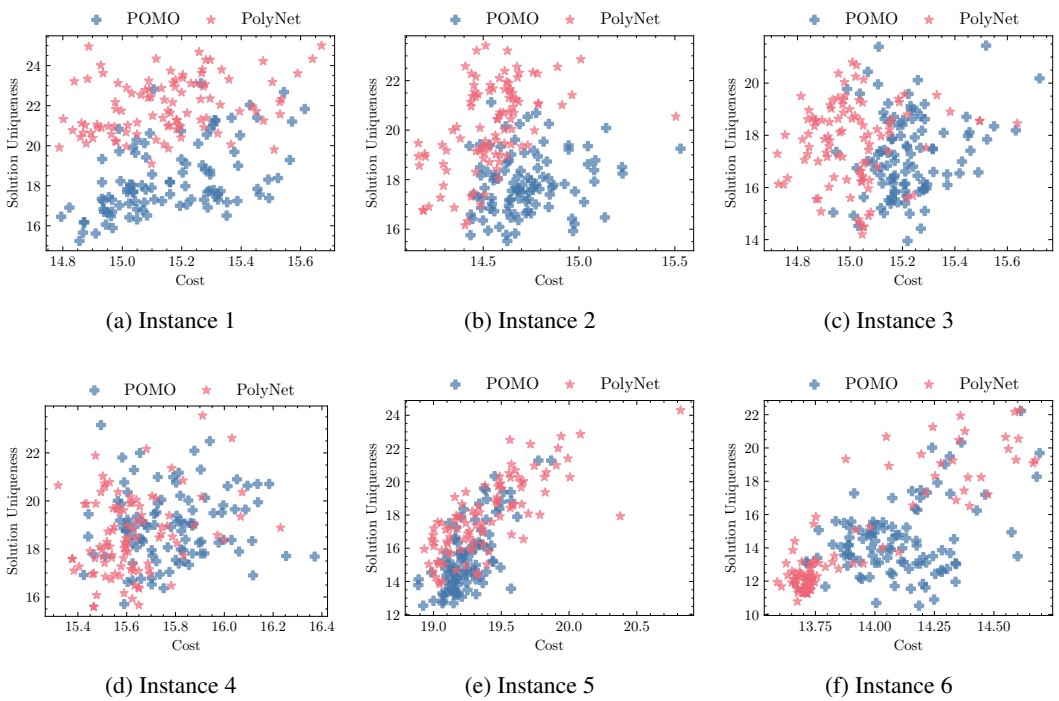

Figure 6: Solution diversity vs. costs for the CVRP

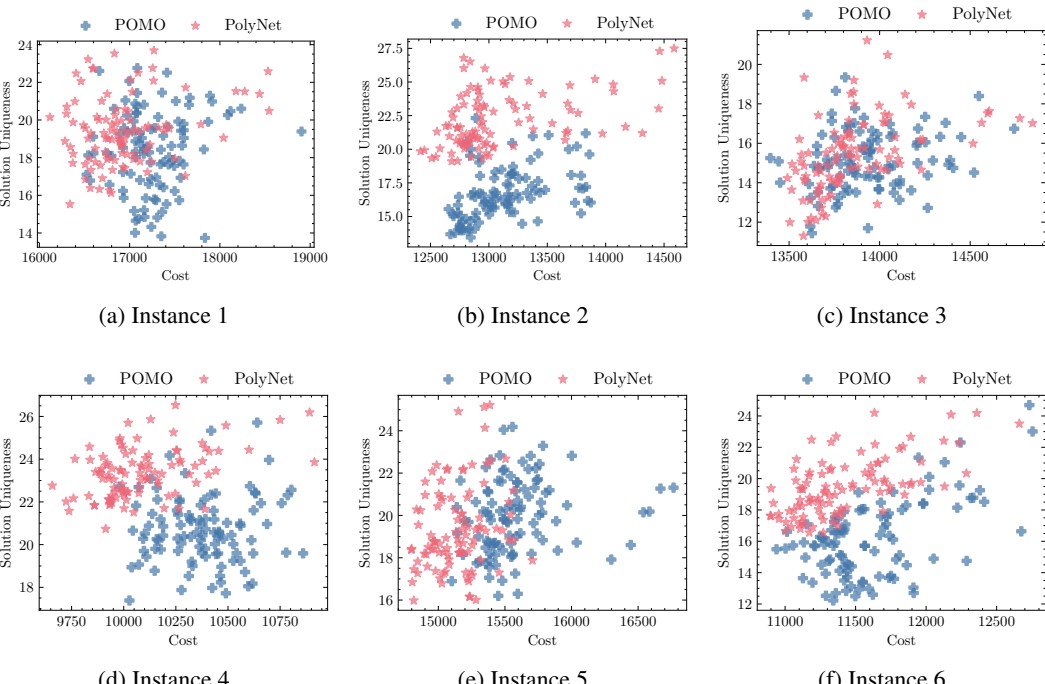

Figure 7: Solution diversity vs. costs for the CVRPTW

## B    SEARCH TRAJECTORY ANALYSIS

In Figure 8 we show the search trajectories for models trained with varying values of $K$ across all three problems featuring 100 nodes. The search process employs solution sampling without EAS and without the use of instance augmentations. These models used for the search have undergone training for 150 epochs (except for the CVRP, where the training spanned 200 epochs).

It is evident across all three problems that the search does not achieve full convergence within 10,000 iterations. This observation once again underscores PolyNet's capability to discover diverse solutions, enabling it to yield improved results with extended search budgets. It's important to note that this experiment, including model training, has not been replicated with multiple seeds. Nevertheless, the results suggest that models trained with larger $K$ values benefit more from longer search budgets compared to models trained with smaller values.

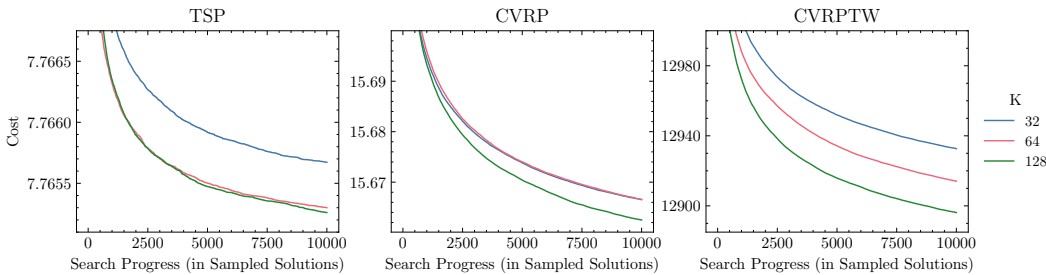

Figure 8: Search trajectories.

## C    CVRPTW INSTANCE GENERATION

We generate instances for the CVRPTW with the goal of including real-world structures. To achieve this, we employ a two-step approach. First, we use the CVRP instance generator developed by Queiroga et al. (2021) to produce the positions and demands of customers. Subsequently, we follow the methodology outlined by Solomon (1987) to create the time windows. We generate different instance sets for training, validation, and testing.

Customer positions are generated using the *clustered* setting (configuration 2) and customer demands are based on the *small, large variance* setting (configuration 2). The depot is always *centered* (configuration 2). It is worth noting that the instance generator samples customer positions within the 2D space defined by $[0, 999]^2$. Independently from the instance generator, vehicle capacities are set at 50 for instances involving fewer than 200 customers and increased to 70 for instances with 200 or more customers.

To generate the time windows, we adhere to the procedure outlined by Solomon (1987) for instances with randomly clustered customers (i.e., we do not utilize the 3-opt technique to create reference routes). We randomly generate time windows $(e_i, l_i)$ for all customers, and set 2400 as the latest possible time for a vehicle to return to the depot. The time window generation process, as described in Solomon (1987), limits the time windows to ensure feasibility (e.g., by selecting $l_i$ so that there is always sufficient time for servicing the customer and returning to the depot). The center of the time window is uniformly sampled from range defined by these limits. We set the maximum width of the time window to 500 and the service duration to 50. These parameter values have been deliberately chosen to strike a balance between the constraints of vehicle capacity and time windows, requiring both aspects to be considered during the solution generation process.

