# OpenReview forum: "PolyNet: Learning Diverse Solution Strategies for Neural Combinatorial Optimization"
_ICLR.cc/2024/Conference — Submitted to ICLR 2024_

### Official Review · Reviewer_eigK · 2023-10-17

**Soundness:** 3 good
**Presentation:** 3 good
**Contribution:** 3 good
**Rating:** 5
**Confidence:** 1

**Summary:**

This paper proposes PolyNet, a novel structure to learn a diverse set of solutions using a single decoder, which is a very light architecture.

**Strengths:**

1. The related work is very well organised and put the paper in a good research position.

**Weaknesses:**

1. Please avoid using vague and less meaningful statements, such as 'has a wider application for ...',especially for the contribution.

2. It is more general put 3 statements in the contribution.

3. t \in (1,..T), this expression is not accurate.

4. The content for describing this new architecture seems not enough. The training process seems standard. In figure 2, it seems polynet only rely on some simple operations on the Bit Vector v. What such a operator will make such a big difference?

5. In section 3.5, the proposed method is combined with an EAS. This makes the unique contribution of this work less clearer.

**Questions:**

1. Why named PolyNet, is anywhere mention it ?

2. The improvement over POMO and others seems very small. Is 0.96% a enough contribution in this field?

3. How is the performance of the proposed method on TSP-10000?

---

> ### Author Response · Authors · 2023-11-20
>
> Thank you for reviewing our paper! Please let us address your concerns and questions:
>
> - **W1**: The statement is not meaningless. Many neural CO methods cannot be used on more than the TSP or some other simple CO problem without significant changes, if they can be used on other problems at all. This work seeks a problem independent method of generating diverse solutions. Even though we focused on routing problems, the mechanism itself is general and requires no modification for other types of CO problems (scheduling, assignment, etc.)
> - **W2**: We are not sure what you mean by that. Can you please elaborate?
> - **W3**: We are not sure what you mean by that. Can you please elaborate?
> - **W4**: Would you prefer it if our method would be more complicated and achieve the same results? We make a number of significant changes to the POMO and PolyNet method that lead to stark performance improvements. Achieving significant performance improvements while maintaining simplicity and efficiency is a commendable achievement.
> - **W5**: We also show the performance of our method without EAS (where it outperforms all other ML methods). We do not see how providing additional results for PolyNet with EAS reduces our contribution. Not all neural CO methods can be combined easily with effective search techniques, thus it is an advantage of PolyNet that EAS can enhance its performance.
>
>
>
>
>
> - **Q1**: This is not mentioned in the paper. “PolyNet” uses the Greek word "poly," which means "many" or "multiple". We chose the name because PolyNet is able to generate multiple diverse solutions.
> - **Q2**: A 1% solution improvement in the area of vehicle routing is very significant as it often translates to tangible cost savings and enhanced resource utilization, contributing to overall efficiency. Even much smaller improvements can provide a competitive advantage.
> - **Q3**: Our approach is not designed for such large problem instances. We refer to our responses to reviewers 2aWJ and 2arF as to why scaling to such large problem sizes is not a valid criteria for judging this work.

---

> > ### Comment · Reviewer_eigK · 2023-11-23
> > **thanks for explanation**
> >
> > Thanks for you explanation. I have no further questions.

---

### Official Review · Reviewer_3QzE · 2023-10-29

**Soundness:** 2 fair
**Presentation:** 2 fair
**Contribution:** 2 fair
**Rating:** 5
**Confidence:** 3

**Summary:**

PolyNet is introduced as an innovative approach that uses a single-decoder model to learn complementary solution strategies for combinatorial optimization problems. It enables the rapid generation of diverse solutions without relying on solution space symmetries, addressing limitations seen in prior reinforcement learning-based methods. Experimental results across different problems consistently show significant improvements in solution generation speed and search effectiveness compared to state-of-the-art machine learning methods.

**Strengths:**

mproving the performance of combinatorial optimization solvers by generating diverse solutions and using them to establish a shared baseline for learning, selecting and utilizing high-performing solutions, has yielded impressive results.

**Weaknesses:**

While it is demonstrated that utilizing a variety of solutions generated by the policy enhances the stability of combinatorial optimization solver learning and improves its final performance, there is a lack of analysis regarding the specific mechanism or reasons for the improvement with the use of K different input vectors.

**Questions:**

1. I'm curious whether the goal is to generate diverse solutions or to train a better-performing solver. If the aim is to obtain diverse solutions, it seems that there is a lack of mechanisms in the learning process to ensure that the K solutions have distinct characteristics.

2. I believe that the approach focuses more on creating K distinct solutions and utilizing them to build a baseline and find the best solution, which is closer to an exploitation strategy for policy learning than generating diverse solutions.

3. I understand that the training uses the best-performing solution among the K solutions, and it seems like the policy learned in this way might tend to overfit to that best solution, improving its performance rather than generating diverse solutions.

4. What is the reason for the effectiveness of the shared baseline composed of K generated solutions?

5. I'm curious about the reason why solution uniqueness is higher in PolyNet compared to POMO. Since there doesn't appear to be a mechanism to explicitly increase uniqueness in the objective function, I wonder if there is causality in the results. I'm also interested in understanding why high uniqueness is desirable when, ultimately, only the best solution is used. Why is it necessary to have diverse solutions?

6. If K vectors are not used, and K solutions are generated using the same policy and the same approach, what would be the resulting outcome?

---

> ### Author Response · Authors · 2023-11-20
>
> Thank you for reviewing our paper! Please let us address your concerns and questions:
>
> - **Q1**: The goal of our approach is to generate better solutions for combinatorial optimization problems. Increasing solution diversity is only a means to an end. We now make this more clear in the paper.
> - **Q2**:  Did you mean to write “exploration strategy”? In that case, you are right.  Our approach creates diverse solutions with the intention to enable wider exploration of strategies and training and test time.
> - **Q3**:  It is not a problem if there is a strategy that is better than all other strategies on a specific instance as long as the other strategies work well on other instances. We only care about the quality of the best found solution (and neither about the quality of the other solutions nor their diversity).
> - **Q4**: The baseline serves the purpose of stabilizing the training process. The adjustment of strategies is not based on the absolute costs achieved, as interpreting these costs can be challenging due to their dependency on both the solution and the instance. Instead, strategies are adjusted relative to how much better they performed compared to other evaluated strategies. This approach allows for a more meaningful and comparative assessment of the strategies, considering their performance in relation to each other rather than focusing solely on absolute cost values.
> - **Q5**:  We are interested in a high-diversity among generated instances because it increases exploration and allows us to find “better best solutions”. If we take a model that offers no solution diversity and generate 800 solutions to a single instance, all solutions will be identical. In contrast, if we use a model with high diversity to generate 800 solutions, all 800 solutions will be different. Assuming that the second model generates not only diverse solutions but also reasonably good solutions, we likely find a better solution in the set of 800 diverse solutions.
> In PolyNet the diversity is increased using the Poppy loss function. The successful implementation of this technique is visually evident in Figure 3 of the paper. When training with PolyNet the solution diversity increases and (more importantly) the quality of the best found solutions increases. We also updated the Figure to include results for continued training of the POMO model to show that POMO does not profit from more training.
> - **Q6**:  Since the policy is conditioned on the additional input vectors, we can not use the same policy without providing K input vectors. Furthermore,the baseline requires a reasonable high value for K (>= 8) during training. At test time, we could sample solutions from only one of the K vectors which would result in decreased solution diversity.

---

> > ### Comment · Reviewer_3QzE · 2023-11-23
> > **Thank you for the responses**
> >
> > It seems that the purpose of the paper has been clearly explained through the rebuttal, and many of my concerns have been addressed. So, I increased my score to 5. However, I still have some reservations as the mechanism for how the diverse set of generated solutions contributes to improving the performance of the final solution is not clearly explained.

---

### Official Review · Reviewer_2arF · 2023-11-01

**Soundness:** 2 fair
**Presentation:** 3 good
**Contribution:** 2 fair
**Rating:** 5
**Confidence:** 5

**Summary:**

The paper proposes the PolyNet method, an enhancement to the constructive solver POMO [1] for solving the VRP (Vehicle Routing Problem). The main contribution is the addition of a residual MLP block in POMO's decoder to produce multiple diverse solutions, which differs from the existing approach Poppy [2] that learns multiple decoders. The approach demonstrates that by using a single decoder, the performance of the POMO model can be enhanced at a relatively small cost. The method has shown state-of-the-art results on small-scale VRP problems.

**Strengths:**

1. A single decoder enhances the diversity of the POMO model at a relatively small cost.
2. Demonstrated state-of-the-art performance on small-scale VRP problems.

**Weaknesses:**

Significance and novelty:

* The technical novelty is slightly limited. The idea of enhancing diversity (similar to EAS [3] and FER [4])  and the integration of additional trainable layers in the POMO decoder (similar to EAS [3] and PMOCO [5]) is not new.
* The addressed TSP and CVRP are relatively small-scale. Many recent works can solve much larger sizes like 10,000 nodes, e.g., DIFUSCO [6].

Evaluation:
* PolyNet added a new residual block to the POMO pre-trained model decoder for further training. However, the baseline models, POMO, Poppy and EAS, did not seem to undergo such extended training. The baseline models should also be trained further for a fair comparison.
Meanwhile, it is unclear how long the additional training is. It is also unclear why PolyNet was not trained from scratch. It's interesting to understand how its performance is when training from scratch.
* It is unclear whether the good performance is attributed to EAS, additional training, or the added layers with bit vector inputs. More evaluation may be needed, e.g., testing PolyNet’s performance over longer inference without EAS, comparing the results of training from scratch and/or additional training (baselines also undergo extended training), and verifying the necessity of the proposed residual block design (e.g., can we change the bit inputs to random values? Can the hyper-network of PMOCO with additional training achieve similar performance?)
* In Figure 4, it seems that both POMO and PolyNet display comparable levels of diversity, though PolyNet found solutions with lower costs. This appears to contradict the primary intention behind PolyNet, which is to amplify diversity. The results raise the question of whether the enhanced performance of PolyNet is due to diversity or if other underlying factors at play contribute to this outcome.

Writing:

The title and some claims have inaccuracies. For instance, the title and other sections mention “applicability to a wider range of CO problems”, but all experiments are conducted only on VRPs. The paper claims that it focuses on practical problems that have many constraints, but the experiments are limited to simple problems like CVRPTW and CVRP. The paper also mentions an "unlimited learnable policy", but the number of decoders is restricted by K.

```
References:
[1] POMO: Policy Optimization with Multiple Optima for Reinforcement Learning
[2] Population-Based Reinforcement Learning for Combinatorial Optimization
[3] Efficient Active Search for Combinatorial Optimization Problems
[4] Learning Feature Embedding Refiner for Solving Vehicle Routing Problems
[5] Pareto Set Learning for Neural Multi-objective Combinatorial Optimization
[6] DIFUSCO: Graph-based Diffusion Solvers for Combinatorial Optimization
```

**Questions:**

Please see my comments above.

---

> ### Author Response · Authors · 2023-11-20
>
> Thank you for reviewing our paper! Please let us address your concerns and questions:
>
> **Significance and novelty**
>
> - The concept of enhancing solution diversity in a search algorithm or inserting new layers into an existing model isn't novel. Indeed, we can cite papers from far earlier than the papers provided in the ML literature. What sets our work apart is our ability to generate varied solutions using a model with a single decoder, showcasing significantly improved performance. This has a number of advantages over the approaches in the ML literature (which rely on multiple decoders) and the OR literature (which rely mostly on randomness). Thus, the novelty is absolutely sufficient for ICLR and we believe our decoder mechanism will have an influence on future autoregressive decoder designs.
>
> - For larger instances of the CVRP and CVRP with Time Windows (CVRPTW), there is currently a lack of end-to-end machine learning approaches. Existing methods, such as DIFUSCO, focus solely on the TSP due to its simpler constraints, enabling non-autoregressive solution generation. In our research, we concentrate on more intricate problems like CVRP and CVRPTW. Even with instances ranging from 100 to 500 nodes, achieving optimal solutions remains extremely challenging, emphasizing the continued importance of developing heuristics for problems of this magnitude. Note that common CVRP benchmarks from the operations research (OR) literature mostly have the same sizes as the problems we examine. This is because size is not the only criteria for the hardness of real-world problems. We encourage the reviewer to orient their thinking on the OR literature, which has been solving these problems for many years. Thus, our work serves to narrow the divide between machine learning and traditional OR methods by orienting the datasets on real problem sizes and structures.
>
> **Evaluation**
>
> - **Additional training of the baseline methods**: In our efforts to enhance baseline methods, Poppy underwent additional training, following the same training protocol as PolyNet, initiated from an already trained POMO model and spanning an identical number of epochs. We have updated our paper to reflect this information. It's worth noting that the POMO model, having been trained to near convergence, does not experience any notable benefits from further training. This assertion is supported by our inclusion of results for additional training in Figure 4. Notably, EAS does not require any training at all. \
> **Training from scratch**: Furthermore, for a comprehensive analysis, we initiated training runs from scratch, and the outcomes of these efforts will be presented in the camera-ready version of our paper.
>
> - **Reason for the good performance**: We have already provided results for PolyNet without EAS in Table 4., affirming that its quality performance is not solely attributed to EAS. Additionally, despite Poppy undergoing identical training to PolyNet, it exhibits inferior performance on identical instances. This observation suggests that PolyNet's superior performance is not merely a result of additional training. Instead the enhanced performance of PolyNet is likely attributable to the novel model architecture and the refined training setup. Importantly, these two components are interdependent, and their effectiveness cannot be evaluated independently.\
> **Comparison to PMOCO**: PMOCO is designed to solve multi-objective optimization problems and is hence not well-suited for the single objective problems considered in this paper.
>
> - **Figure 4**: PolyNet does not only focus on solution diversity, but aims to increase solution diversity and solution quality alike. Figure 4 shows the trade-off between these two objectives. Overall, PolyNet finds solutions that are better than those of POMO with respect to these two objectives (also see Appendix A). To allow for a direct comparison of the solution diversity of POMO vs. PolyNet, we updated Figure 3 to include POMO.
>
> **Writing**
>
> - **Considered problems**: We are currently working on including results for the flexible flow shop problem. The results will be included in the camera-ready version of the paper.
> - **Practical problems**: The CVRPTW in particular is significantly more complex than most other problems considered in the neural combinatorial optimization literature. We hence see our work as an important step to more complex problems.
> - **“Unlimited learnable policy”**: We rephrased this statement to make it more clear.

---

> > ### Comment · Reviewer_2arF · 2023-11-22
> >
> > I thank the authors for their response and have taken into account the comments from other reviewers.
> >
> > While the authors have addressed some of my concerns, their response still lacks experimental evidence, and the promised additional results in the final version may not address the current concerns. The paper demonstrates promising performance with simple designs, which could have a positive impact. However, the paper currently falls short in providing a thorough evaluation and discussion. It particularly lacks depth in explaining the performance origins of PolyNet, the rationales and unique benifits behind the proposed residual block designs, and whether the performance improvements are due to increased diversity. . For these reasons, I will maintain my initial review score.

---

> > > ### Author Response · Authors · 2023-11-22
> > >
> > > You are of course under no obligation to adjust your score. We are disappointed that we did not receive more concrete explanation as to what experiments are missing so that we can improve our paper.
> > >
> > > We believe our experiments show rather convincingly that PolyNet’s good performance is due to the added layers, and not EAS. Furthermore, our experiments are in line with previous papers at NeurIPS and ICLR, e.g., POMO and EAS.

---

> > > > ### Comment · Reviewer_2arF · 2023-11-23
> > > > **Further clarifications**
> > > >
> > > > I acknowledge that the good performance is not solely due to EAS. However, my main concern remains the lack of thorough discussion and evaluation to substantiate the benefits of the proposed design changes. The assertion that the added layer and refined training setup are interdependent, leaves me unconvinced. In my initial review, I recommended further evaluations to ascertain the unique benefits and necessity of the newly added layer. Possible ways include replacing the residual block with PMOCO's hyper network, using random bit inputs instead of one-hot coding, or applying the introduced layers in the EAS-lay as a trainable layer atop a pre-trained POMO model. Subsequently, one could conduct similar additional training to determine whether the design of the residual block or the additional training itself contributes more significantly to the model's enhanced performance.
> > > >
> > > > There is also several unanswered questions:
> > > >
> > > > - What is PolyNet's performance when operating independently, especially over longer inference periods (consider the simialr runtime ofthe guided group's), without EAS?
> > > > - What is the total wall time required for the additional training?
> > > > - How does PolyNet perform when trained from scratch? Any preliminary insights?
> > > >
> > > > Additionally, upon a carefully re-evaluation of the paper's contributions adn check the experiments of POMO and EAS, I acknowledge the admirable performance gains. Should my above concerns be adequately addressed, I am happy  to reconsider and increase my score.

---

### Official Review · Reviewer_2aWJ · 2023-11-09

**Soundness:** 3 good
**Presentation:** 3 good
**Contribution:** 3 good
**Rating:** 5
**Confidence:** 3

**Summary:**

The paper presents a new learning-based approach to solving the combinatorial optimisation problem by using the single-decoder model to improve the diversity mechanism. The experiments on three selective combinatorial optimisation problems showed that there is an improvement over the other method.

**Strengths:**

Originality: The new learning-based method employed a single-decoder model to improve the diversification of the exploration.

Quality: The overall quality is good. The authors performed substantial experiments with other methods on three categories of testing instances TSP, CVRP, CVRPTW

Clarity: The paper is generally well-written, and the overall clarity is good.

Significance: The experiments were substantial and significantly showed the improvement of the method over the other methods. However, I would like to see more experiments of PolyNet with SGBS, and PolyNet with SGBS+EAA. There is also a need to have a better experiment in part 4.5 with Poppy free first move and Poppy forced first move.

**Weaknesses:**

The author already admitted the high computational complexity of the methods in the conclusion.
It seems to me that the PolyNet is not the fastest solver in Table 1, 2, 3.

**Questions:**

On table 1, there is no information about the performance o PolyNet EAS on Test 10K instances.
In Table 4, does the poppy method have the forced first move or free first move?
For part 3.4, can you explain more about how the diversity can be increased via the loss function?
Part 3.5: Can you clarify more “This approach generates a diverse set of instances in a parallel and independent manner, making it particularly suitable for real-world decision support settings where little time is available?”. Can we say that the selected testing instances are represented for real-world decision support?
Part 3.5: Why did not you combine PolyNet with SGBS?

---

> ### Author Response · Authors · 2023-11-20
>
> Thank you for reviewing our paper! Please let us address your concerns and questions:
>
> **High computational complexity**: We solve problems of sizes relevant for many real-world contexts. Let us emphasize that the standard datasets in Operations Research for the CVRP, etc., also focus on problems of this size because this is what the vast majority of companies are interested in. As this work is trying to move the field of neural CO towards real-world impact, it makes sense that we would focus on these problems. Note that we are open about the scaling limitation of this work and state in the discussion that the computational complexity of our method limits its applicability to routing problems with less than 1,000 nodes.
>
> **Performance**: PolyNet outperforms all other state-of-the-art approaches for the routing problems we consider. In Tables 1, 2, and 3, PolyNet is faster than all other approaches except for POMO with greedy action selection (and POMO with sampling in one case). In addition to being faster, PolyNet finds significantly better solutions for all three problems.
>
>
>
> **Q1**: “On table 1, there is no information about the performance o [sic] PolyNet EAS on Test 10K instances.”
>
> Yes, we did not conduct these experiments because PolyNet without EAS already finds solutions with a gap of 0.000% to optimality and it is hence impossible to meaningfully improve the solution quality further. However, we agree that the missing values might be confusing, and we now include results for these experiments in the paper.
>
> **Q2**: “In Table 4, does the poppy method have the forced first move or free first move?”
>
> We implement Poppy as proposed by the authors without any modifications. This means that for Poppy the first move is always forced. We note that Poppy (similar to POMO) has been designed around this mechanism and that replacing it with our free first move selection would require extensive changes to the method.
>
> **Q3**: “For part 3.4, can you explain more about how the diversity can be increased via the loss function?”
>
> In our approach, we implement the diversity-enhancing mechanism explained in the Poppy paper. Given the page limit constraint, we present a concise overview of this mechanism and direct readers to the detailed exposition in the Poppy paper. Our strategy for fostering diversity involves sampling K solutions per instance and subsequently updating the model weights based on the best sampled solutions. While this approach doesn't explicitly enforce diversity, it serves as an incentive for the model to acquire varied strategies in order to optimize overall performance. It's crucial to emphasize that our pursuit of diversity is fundamentally a means to enhance solution quality, a goal we have effectively achieved. To enhance clarity, we have revised accordingly.
>
> **Q4**: “Can you clarify more “This approach generates a diverse set of instances in a parallel and independent manner, making it particularly suitable for real-world decision support settings where little time is available?”. Can we say that the selected testing instances are represented for real-world decision support?”
>
> In the case of CVRPTW, we have taken great care to have a real-world relevant dataset. We employ the CVRP instance generator detailed by Queiroga et al. (2021). This generator produces clustered customer positions that closely emulate real-world scenarios, introducing a layer of realism to our evaluation. In essence, while our instances may not directly mirror real-world problems, the structural design of our CVRPTW instances aligns with those encountered in practical scenarios and corresponds to instances commonly employed in the operations research literature.
>
> In the case of the TSP and CVRP, we evaluate PolyNet on simple instances, as these problems are not real world anyway. In the case of TSP, the node positions are randomly sampled from a uniform distribution, facilitating seamless comparisons with other machine learning papers.
>
> **Q5**: “Part 3.5: Why did not you combine PolyNet with SGBS?”
>
> Integrating PolyNet with SGBS is non-trivial due to the way PolyNet generates solutions and SGBS performs its search. We made a first attempt at integrating the two methods, but are not yet satisfied with its performance and have thus delayed the task of fully integrating the two methods for future work.

---

> > ### Comment · Reviewer_2aWJ · 2023-11-23
> > **Thank you**
> >
> > Thank you authors for addressing the questions. I do not have any further questions.

---

### Meta-Review · Area_Chair_tt4G · 2023-12-05

**Metareview:**

The paper attempts to have more diverse proposals during RL-based combinatorial optimization, by representing a set of diverse behavioral policies using a single decoder but prompted with different bitstrings, called "PolyNet". Training was done via a "diverse best-so-far policy gradient" (Equation 2), similar to Poppy [1].

Comparisons were performed extensively against [1] and also POMO [2], a method which performs policy gradients over symmetry-augmented replay data.

Experiments were conducted over travelling salesman and capacity-bounded variant problems, with results:
* Lower-cost solutions found by PolyNet
* More diverse solutions than POMO

During the review process, there were numerous questions about the proposed method, e.g.
* Does PolyNet work on non-routing based problems, or on larger scales?
* What parts of the entire PolyNet training experiment actually contributes to better performance / was a fair comparison done?
* Why does diversity contribute to better performance, and how does PolyNet improve diversity despite lack of an explicit diversity objective?

Due to these multiple concerns, some addressed during the rebuttal phase but some still unaddressed, for now the paper should take these issues into consideration and resubmit to another venue.

[1] https://arxiv.org/abs/2210.03475

[2] https://arxiv.org/abs/2010.16011

**Justification For Why Not Higher Score:**

Reviewers raised numerous concerns about the paper, some were still unaddressed even after the rebuttal phase (e.g. performance on non-routing problems, ablation studies detailing why PolyNet exactly works).

**Justification For Why Not Lower Score:**

N/A

---

### Decision · Program_Chairs · 2024-01-16

Reject